# Adaptive Phage Therapy for the Prevention of Recurrent Nosocomial Pneumonia: Novel Protocol Description and Case Series

**DOI:** 10.3390/antibiotics12121734

**Published:** 2023-12-14

**Authors:** Fedor Zurabov, Marina Petrova, Alexander Zurabov, Marina Gurkova, Petr Polyakov, Dmitriy Cheboksarov, Ekaterina Chernevskaya, Mikhail Yuryev, Valentina Popova, Artem Kuzovlev, Alexey Yakovlev, Andrey Grechko

**Affiliations:** 1Research and Production Center “MicroMir”, 107031 Moscow, Russia; azurabov@micromir.bio (A.Z.); mgurkova@micromir.bio (M.G.); val.popova@micromir.bio (V.P.); 2Federal Research and Clinical Center of Intensive Care Medicine and Rehabilitology, 10703 Moscow, Russia; mpetrova@fnkcrr.ru (M.P.); p.polyakov@fnkcrr.ru (P.P.); dcheboksarov@fnkcrr.ru (D.C.); echernevskaya@fnkcrr.ru (E.C.); myurev@fnkcrr.ru (M.Y.); artem_kuzovlev@fnkcrr.ru (A.K.); ayakovlev@fnkcrr.ru (A.Y.); avgrechko@fnkcrr.ru (A.G.)

**Keywords:** phage therapy, bacteriophages, antimicrobial resistance, regulatory framework, personalized medicine, clinical case, intensive care

## Abstract

Nowadays there is a growing interest worldwide in using bacteriophages for therapeutic purposes to combat antibiotic-resistant bacterial strains, driven by the increasing ineffectiveness of drugs against bacterial infections. Despite this fact, no novel commercially available therapeutic phage products have been developed in the last two decades, as it is extremely difficult to register them under the current legal regulations. This paper presents a description of the interaction between a bacteriophage manufacturer and a clinical institution, the specificity of which is the selection of bacteriophages not for an individual patient, but for the entire spectrum of bacteria circulating in the intensive care unit with continuous clinical and microbiological monitoring of efficacy. The study presents the description of three clinical cases of patients who received bacteriophage complex via inhalation for 28 days according to the protocol without antibiotic use throughout the period. No adverse effects were observed and the elimination of multidrug-resistant microorganisms from the bronchoalveolar lavage contents was detected in all patients. A decrease in such inflammatory markers as C-reactive protein (CRP) and procalcitonin was also noted. The obtained results demonstrate the potential of an adaptive phage therapy protocol in intensive care units for reducing the amount of antibiotics used and preserving their efficacy.

## 1. Introduction

One of the challenges of the 21st century is the ineffectiveness of drugs against bacterial infections [1]. Hospital-acquired infections currently cause millions of deaths per year, and the prognosis for the future is even worse, as evidenced by current research. In 2009, over 20,000 people in the US died due to the lack of effective antibiotics. The same statistics have been observed in Europe. Worldwide, more than 100,000 people have died from infections caused by antibiotic-resistant bacteria [2]. In 2019, the annual number of deaths associated with antibiotic-resistant bacteria surged to 4.95 million [3].

Phage therapy is based on the therapeutic use of bacteriophages to treat bacterial infections. Since Felix D’Hérelle’s introduction of the term “bacteriophage” in 1917, phages have been used extensively in combatting bacterial pathogens. However, in Western Europe and the USA, phage therapy was soon abandoned due to questionable treatment outcomes, lack of standardization, and the discovery of antibiotics. Bacteriophages continued to be used for therapy in Eastern European nations like the USSR, supported by the establishment of the Bacteriophage Institute in Tbilisi in 1934 by George Eliava together with Felix D’Hérelle [4]. A reduction in treatment options for patients due to the spread of antibiotic resistance among bacterial pathogens has led researchers to explore new methods of managing bacterial populations and restoring the natural balance of the microbiota. Currently, there is a growing interest worldwide in using bacteriophages for therapeutic purposes to combat antibiotic-resistant bacterial strains, and they are increasingly being used by scientists and clinicians [5]. Published clinical cases and individual clinical trials of phage therapy suggest a high level of safety when using different routes of administration in humans, although some studies do not prove efficacy [6]. Humans co-exist with phages throughout their lives, and they are an inherited element of the human microbiome [7], so the occurrence of unwanted side effects, including toxicity or allergy, is highly unlikely. Furthermore, phage therapy has proven to have practically no side effects [8].

In spite of their importance, no new commercially available therapeutic phage products have been developed in the last two decades. This is largely due to the regulatory restrictions that exist for bacteriophages, which make it impossible to register these products under current legal frameworks [9]. In both Europe and the USA, there is no specific regulation of bacteriophages. Since 2011, phages have been classified as a drug in the USA or as a medicinal product in the European Union [10]. The qualification of phages as a medicinal product for human use was approved in 2015 at a workshop of the European Medicines Agency (EMA), despite warnings from phage researchers about the inadequacy of such an assessment. Following the event, researchers expressed their disagreement in a letter highlighting the need for a novel regulatory structure [11].

Clinical trials of phage products have revealed a range of problems associated with double-blind studies, which are costly and time-consuming for the testing of at least part of the drug formulation [2]. Additionally, a significant downside of commercializing phage-based products is the requirement to create phage cocktails that target the largest number of strains of the given bacterium. Furthermore, these cocktails require regular updates to ensure they are effective against currently circulating clinical strains. Unfortunately, this strategy cannot be implemented due to current drug approval regulations [9]. Nonetheless, studies indicate that authorities responsible for licensing phage therapy treatments in Europe and the United States are attempting to streamline the licensing process [2].

It is important to be able to use newly discovered phages quickly. One way to do this would be through the reduction of licensing requirements to the definition of a full production cycle. There is ongoing discussion regarding possible licensing pathways for phage products. A licensing pathway model should be created with consideration given to the fact that bacteriophages are natural, biological agents that can specifically combat pathogenic bacteria, while self-regulating [2]. Researchers suggest that the licensing process for phage therapy should be adapted to its unique characteristics, rather than the other way around [12].

Because of the aforementioned difficulties in registering bacteriophage products, they are often used within the confines of Article 37 of the Declaration of Helsinki [13]. Article 37 of the Declaration of Helsinki states that physicians may use an unproven intervention for an individual patient only when proven or known methods are ineffective, the physician has sought expert advice, has obtained informed consent from the patient or the patient’s legal representative, and believes that the unproven method offers hope of saving the life, restoring the health, or alleviating the suffering of the patient. According to Pirnay et al. [14], only 28 patients were treated with personalized phage therapy between 2018 and 2022 (based on published case reports and case series). These sparse patient numbers suggest that the personalized approach to phage therapy, although effective in many cases, is difficult to scale up and does not provide access to phage therapy for a wide range of patients.

The most significant advancement in regulating phage therapy took place in Belgium, where in 2017, the government, under the guidance of a team of researchers from the Queen Astrid Military Hospital in Brussels, decided to classify phages not as industrial drugs, but as active components in magistral preparations. A “magistral preparation” is defined as “any medicinal product prepared in a pharmacy according to a physician’s prescription for an individual patient” [15]. This process enabled a Belgian hospital to produce phages for the treatment of bacterial infections in humans. However, there are still questions that need to be clarified, as there is no clear consensus on the requirements and standards for the production of such preparations.

Scientists and clinicians in the Russian Federation are developing a comparable approach to legalize the administration of bacteriophages. The difference is that a “cocktail” of bacteriophages is prepared not for an individual patient but for a specific medical institution/department based on the results of bacterial composition monitoring, which is called “adaptive phage therapy” [16]. In order to expand and legitimize the approach, pharmaceutical substances must be registered and manufactured into final forms by compounding pharmacies. The first pharmaceutical substance, “Bacteriophages specific to *Klebsiella pneumoniae*”, was registered in July 2023 by the Research and Production Centre (RPC) “Micromir”.

The present article focuses on the description of the adaptive phage therapy protocol for patients in intensive care units, the specificity of which is the selection of bacteriophages not for an individual patient, but for the entire spectrum of bacteria circulating in the intensive care unit with continuous clinical and microbiological monitoring of efficacy, and the description of case series of patients who received bacteriophage complex via inhalation for 28 days according to the protocol without antibiotic use throughout the period.

## 2. Results

### 2.1. Adaptive Phage Therapy Protocol

The protocol was created to describe the interaction between the Research and Production Centre (RPC) “Micromir” acting as a phage center and a production site certified under the rules of GMP, and the intensive care units (ICU) of the Federal Research and Clinical Center of Intensive Care Medicine and Rehabilitology (FRCC ICMR). The main goal is to start using an adapted bacteriophage cocktail in all patients admitted to the ICU to prevent and treat nosocomial pneumonia. The interaction cycle is continuous with time constraints on the execution of every stage. A schematic description of the protocol is presented in Figure 1.

FRCC ICMR collects clinical material samples from all patients and isolates pure cultures with a minimum of 50 isolates of each target bacterial species. The period of sample collection is 30 days. The research and production center (RPC) “Micromir” conducts sensitivity testing of pure cultures to bacteriophages from its phage bank with an execution period of up to 21 days. After that the RPC “Micromir” produces an adapted phage cocktail with an efficacy of at least 70% to each species of target bacteria and transfers the adapted cocktail to the FRCC ICMR, where it is applied in intensive care units 2–3 times a day in a dose of 5 mL with a nebulizer.

Clinical evaluation of the efficacy of the adapted bacteriophage complex application in patients is carried out by FRCC ICMR, which analyses lung CT scans, blood tests, clinical data, and microbiological and PCR tests of bronchoalveolar fluid and summarizes the results once a month. To verify the sensitivity of cultures to the adapted complex, FRCC ICMR weekly transfers up to five samples of pure cultures to the laboratory of RPC “Micromir” for sensitivity tests of bacterial isolates to the adapted bacteriophage cocktail. Based on the results of monitoring in one time interval (1 month), if clinical efficacy is absent in more than 50% of patients and/or less than 60% of each bacterial species isolates are sensitive to the current adapted phage cocktail, a decision to start a new cycle of adaptation is made.

### 2.2. Adapted Phage Cocktail Application

This section presents the first clinical cases of intensive care unit patients who received adaptive phage therapy without the need for antibiotics.

1. Patient 92/23, 42 years old, was admitted to the intensive care unit of FRCC ICMR from another medical center. Based on the patient’s history, antimicrobials (AMPs) were administered to treat hospital-acquired pneumonia, urological infection, and consequent colitis. According to the data of computed tomography of the chest organs on the day of admission, the presence of focal masses and infiltrative changes in the lungs were not detected. A CT scan shows hypostatic and post-inflammatory changes in the right lung. These changes are indicative of a previous case of pneumonia. Clinical parameters of the patient are presented in Table 1.

The results of PCR-diagnostics on the day of admission revealed the presence of multidrug-resistant *A. baumannii* (10^5^ CFU/mL) in bronchoalveolar lavage and multidrug-resistant (except for Trimethoprim and Fosfomycin) *K. pneumoniae* (10^5^ CFU/mL). No other pathogenic bacteria were detected in the investigated samples. The results of sensitivity testing of the isolated bacteria to antibiotics and the detection of resistance markers are presented in Appendix A.

The effectiveness of complex phage therapy is confirmed by the data of microbiological studies and laboratory-instrumental data. No growth of pathological bacteria was detected on the 28th day. According to computed tomography data, no evidence of pneumonia is detected.

During the whole period of treatment and rehabilitation measures in the intensive care unit, no cases of infectious and septic complications were observed in the patient, which fully allowed to implement the range of rehabilitation measures and qualitatively improve the patient’s condition. 

Precisely 28 days after admission, the course of therapeutic and rehabilitative measures in the ICU was completed, and the patient was transferred to the neurorehabilitation department with improvement for further therapy. No antibiotics were administered to the patient throughout the stay in the ICU.

2. Patient 62/22, 59 years old, was admitted to the intensive care unit of FRCC ICMR from another medical center. Based on history, the patient was prescribed antimicrobials (AMPs) to treat hospital-acquired pneumonia and colitis. Computed tomography of the chest organs at the time of admission indicates inflammatory-atelectatic changes in the lower lobe of the right lung and post-inflammatory changes in both lungs. Clinical parameters of the patient are presented in Table 2.

The results of PCR-diagnostics on the day of admission revealed the presence of multidrug-resistant *A. baumannii* (10^4^ CFU/mL), *K. pneumoniae* (10^5^ CFU/mL), *P. aeruginosa* (10^5^ CFU/mL), and *S. aureus* (10^5^ CFU/mL) in bronchoalveolar lavage. No other pathogenic bacteria were detected in the investigated samples. The results of sensitivity testing of the isolated bacteria to antibiotics and the detection of resistance markers are presented in Appendix A.

The effectiveness of complex phage therapy is confirmed by the data of microbiological studies and laboratory-instrumental data. No growth of pathological bacteria was detected on the 28th day. According to computed tomography data, improvement of bronchial patency on the right side is noted.

During the whole period of treatment and rehabilitation measures in the intensive care unit, no cases of infectious and septic complications were observed in the patient, which fully allowed the implementation of the range of rehabilitation measures to qualitatively improve the patient’s condition. 

Precisely 28 days after admission, the course of therapeutic and rehabilitative measures in the ICU was completed, and the patient was transferred to the neurorehabilitation department with improvement for further therapy. No antibiotics were administered to the patient throughout the stay in the ICU.

3. Patient 847/21, 73 years old, was admitted to the intensive care unit of FRCC ICMR from another medical center. Based on history, the patient was prescribed antimicrobials (AMPs) to treat hospital-acquired pneumonia. Computed tomography of the chest organs at the time of admission indicates bilateral lower lobe pneumonia. Clinical parameters of the patient are presented in Table 3.

The results of PCR-diagnostics on the day of admission revealed the presence of multidrug-resistant *A. baumannii* (10^5^ CFU/mL) and *K. pneumoniae* (10^4^ CFU/mL) in bronchoalveolar lavage. No other pathogenic bacteria were detected in the investigated samples. The results of sensitivity testing of the isolated bacteria to antibiotics and the detection of resistance markers are presented in Appendix A.

The effectiveness of complex phage therapy is confirmed by the data of microbiological studies and laboratory-instrumental data. No growth of pathological bacteria was detected on the 28th day. According to computed tomography data, in the posterior sections of the lower lobes of both lungs, the areas of “ground glass” in combination with reticular changes are preserved; positive dynamics in the form of reduction in size and severity is noted. A decrease in the extent and density of “ground glass” type areas and consolidation in the upper lobe of the right lung is noted; in the upper lobe of the left lung a single focal area of ground glass type is preserved, positive dynamics is noted.

Precisely 28 days after admission, the course of therapeutic and rehabilitation measures in the ICU was completed. For further therapy the patient was transferred with improvement to the palliative care unit, where no further infectious and septic complications were observed. No antibiotics were administered to the patient throughout the stay in the ICU.

## 3. Discussion

Currently, the use of phages in healthcare facilities is mainly limited to individualized selection of bacteriophages for each patient suffering from an antibiotic-resistant infection [2]. This process is very challenging to scale up, leading to a loss of time that can be life-threatening for some patients. Magistral phage production in Belgium is a more convenient process as it allows phages to be selected from stock banks of pre-purified bacteriophage lines, then transferred to suitable GMP manufacturing sites and then to physicians for use according to prescription in individual patients [15].

The proposed adaptive phage therapy technology implies strict compliance of a set of bacteriophages to the needs of a particular intensive care unit rather than a particular patient. This avoids the necessity of individual bacteriophage selection, as the cocktail is prepared based on sensitivity testing of bacteria obtained from many patients of the same ICU. This reduces the decision time required for immediate initiation of therapy, which will help to improve the effectiveness of treatment for critically ill patients. We aim for this technology to decrease the usage of antibiotics in ICUs and enhance their efficacy when required. Studies indicate that phage application can, in certain cases, restore the susceptibility of bacterial strains to antibiotics, as antibiotic resistance mechanisms can be lost in the process of bacterial population adaptation to phage infection [17]. Moreover, in the conducted in vivo study, the combination of phage and antibiotic has demonstrated a higher bactericidal effect against severe *A. baumannii* infection, compared to each agent individually. Phage øFG02 has been shown to consistently stimulate the in vivo evolution of *A. baumannii* towards a capsule-deficient, phage-resistant phenotype that became sensitive to ceftazidime [18]. This mechanism highlights the clinical potential of phage therapy in combination with antimicrobial therapy for restoring antimicrobial activity and reducing the amount of antibiotics used.

In the present study, the cocktail of bacteriophages for inhalation including 3–4 virulent bacteriophages to each bacterial species was administered to patients, as it is recommended to use multiple phages in therapy to prevent the development of bacterial resistance to phages as well as to expand the phage-host range and increase the number of target pathogens [19]. In some clinical cases, researchers have noted that the use of several phages simultaneously may negatively affect the efficacy of individual phages, but detailed data on antagonism were not presented, and the primary factor contributing to the negative clinical outcome was multi-organ failure [20].

All patients included in this study received a cocktail of bacteriophages according to one regimen: 5.0 mL of the solution per inhalation two times a day for 28 days. It is noteworthy that patients did not receive any antibiotics during the whole course of phage cocktail administration. During the therapy, no adverse events and side effects were reported. Instead, all patients showed improvement on day 28 of the therapy. The primary outcome was the elimination of multidrug-resistant microorganisms (*K. pneumoniae*, *S. aureus*, *A. baumannii*) from the BAL contents in all patients. Following bacterial elimination, all patients demonstrated positive dynamics according to lung computed tomography data; 2/3 of patients also showed improvement in nature of bronchial secretion according to Clinical Pulmonary Infection Score (CPIS). Moreover, a decrease in such inflammatory markers as CRP was observed in all patients. In the case of initially increased level of procalcitonin, there was also a decrease in this indicator. A retrospective analysis of bacteriophage administration in 37 patients also showed a significant decrease in mean CRP values measured between days 9 and 32 [21]. This may be due to a reduction in the intensity of the inflammatory response due to a decrease in bacterial load. In conducted studies, a decrease in CRP levels during bacteriophage administration was also noted [16,22].

The major limitation of this study is the small sample size to ascertain the statistical validity of the obtained results. Case reports are generally not the basis for testing statistical hypotheses but are used to create hypotheses for future research. The main purpose of this manuscript is to describe the protocol of adaptive phage therapy and individual clinical experience. The findings suggest the potency of adaptive phage therapy and the feasibility of extending the study to larger groups. Despite the demonstrated efficacy of adaptive phage therapy, we do not exclude cases where in some individuals this approach will result in a lack of clinical efficacy, as bacteriophages are not selected for the individual patient. Therefore, this approach involves close cooperation between the clinical institution and the bacteriophage manufacturer, regular clinical monitoring, sensitivity testing of bacterial strains to the action of bacteriophages, and adaptation of the phage cocktail. The main advantage of the adaptive phage therapy approach is the possibility to start phage cocktail administration from the first days of the patient’s admission to the ICU.

An additional limiting factor was the exclusive use of EUKAST standards in the evaluation of sensitivity to protected β-lactam antibiotics (Amoxicillin/Clavulanate). The use of CLSI standards will enable the evaluation of different antimicrobial to beta-lactamase inhibitor ratios in future studies. It has been demonstrated that the CLSI and EUCAST methodologies showed poor concordance in determining the MIC of amoxicillin/clavulanate [23]. MIC values obtained using the EUCAST methodology were more predictive of failure than those obtained using the CLSI methodology. EUCAST-derived MIC values >16/2 mg/L were independently associated with therapeutic failure. The described method may be a promising way to reduce the amount of administered antibiotics and maintain their efficacy. It appears to be more convenient and faster than classical individual phage therapy. However, scaling up this approach may cause some difficulties, as it requires the shipment of pure cultures from the health care facility to the phage center. Regrettably, not all medical facilities have sufficient medical personnel and equipment to qualitatively isolate and characterize pure bacterial cultures. Moreover, sending materials over long distances also involves logistical difficulties and safety risks. The solution may be the establishment of a network of phage centers and authorized laboratories that will work in cooperation with medical institutions. For large-scale application of adaptive phage therapy without the need for approval of the ethical committee of each individual hospital, registration of a sufficient number of phage pharmaceutical substances is essential.

The conducted study demonstrates the potential of an adaptive phage therapy protocol in intensive care units. We will continue to investigate this method and its impact on the quality of patient care, as well as on the amount of antibiotics used in the ICU and their efficacy. 

## 4. Materials and Methods

### 4.1. Participants

All patients at the time of inclusion in the study had no clinical, laboratory, or instrumental signs of systemic inflammatory complications requiring the prescription of antimicrobial drugs. Patients did not receive antibiotic therapy during the 28-day stay. Treatment and rehabilitation measures were performed by specialists who had no information about the inclusion of patients in this study. Patients were selected randomly according to the following criteria.

Inclusion criteria: Patient age >18 years;Chronic critical condition;Absence of acute systemic infection requiring the use of antimicrobial therapy (AMT) at the time of hospitalization in Federal Research and Clinical Center of Intensive Care Medicine and Rehabilitology (FRCC ICMR);Antimicrobial therapy at the previous stages of hospitalization;Informed consent from the patient or next of kin for inclusion in the study.

Exclusion criteria:Low chance of survival, Simplified Acute Physiology Score (SAPS) II score of more than 65;Treatment with immunosuppressants or corticosteroids;Oncological diseases;Evidence of systemic severe infection (Sepsis-3 criteria);Candidemia.

Three patients were included in the present study: Patient 92/23, female, 42 years old; Patient 62/22, male, 59 years old; Patient 847/21, male, 73 years old. Prior to admission to FRCC ICMR, Patient 92/23 underwent treatment with the diagnosis: consequences of subarachnoid haemorrhage from a saccular aneurysm of the supraclinoid aneurysm of the right internal carotid artery. She underwent surgery: pterional craniotomy on the right side, clipping of the supraclinoid aneurysm of the right internal carotid artery. Patient 62/22 was treated for intracerebral haemorrhage in the left thalamus region prior to admission to FRCC ICMR. The course of the disease was complicated by the development of occlusive deep vein thrombosis on the right side, hospital-acquired lower lobe pneumonia, and multi-organ failure. A puncture-dilatation tracheostomy was performed. Prior to admission to FRCC ICMR, patient 847/21 was treated for subcortical haemorrhage in the left cerebral hemisphere with blood breakthrough into the liquor system. The course of the disease was complicated by the development of hospital-acquired lower lobe pneumonia; puncture-dilatation tracheostomy was performed.

Patients underwent a set of therapeutic and rehabilitation measures: maintenance of functions of vital organs and systems, pharmacological correction of the level of consciousness, nutritional and metabolic therapy, symptomatic treatment. Acid-base status analyses, including PaO2 analysis, were taken every 7 days to assess the severity of the patients’ condition. The analyses were performed using a GemPremier 3500 analyzer (Version 7.2.5, Instrumentation Laboratory, Bedford, MA, USA).

To prevent recurrence of nosocomial pneumonia, patients received an adapted complex of bacteriophages. Phage therapy was carried out from the first day of the patient’s admission to the intensive care unit (ICU) of FRCC ICMR by aerosol therapy using a nebulizer with 5.0 mL of the solution per inhalation 2 times a day.

The cocktail for inhalation included 3–4 virulent phages to each bacterial species active against clinical strains of *Acinetobacter baumannii*, *Stenotrophomonas maltophilia*, *K. pneumoniae*, *K. pneumoniae subsp. ozanae*, *Pseudomonas aeruginosa*, *Staphylococcus aureus*, *Staphylococcus epidermidis*, *Staphylococcus warneri*, *Staphylococcus haemolyticus*, *Staphylococcus capitis*, *Staphylococcus caprae*, *Staphylococcus succinus*, *Streptococcus pyogenes*, *Streptococcus agalactiae*. Each bacteriophage was produced in a separate production and purification series in accordance with good manufacturing practice (GMP) standards. The preparation consisted of a sterile suspension of phage particles in a physiological solution. The titer of each bacteriophage was 10^5^–10^6^ PFU/mL.

### 4.2. Clinical Monitoring

Patients were under constant clinical and laboratory monitoring with evaluation of indicators of the cardiovascular system, neurological status, respiratory, liver, kidney function, and the level of organ dysfunction. The levels of inflammatory biomarkers in serum (C-reactive protein, procalcitonin) were measured in dynamics. The determination of transferrin and C-reactive protein (CRP) levels was performed on an automatic bio-chemical analyzer AU 480 (Beckman Coulter, Brea, CA, USA) using original reagents. Procalcitonin level was determined on immunological analyzer VIDAS (bioMerieux SA, Marcy-l’Étoile, Lyon, France). General clinical blood parameters (leukocytes, neutrophils, platelets, lymphocytes) were determined on an automatic hematological analyzer Sysmex XN550 (Sysmex, Kobe, Hyogo, Japan). 

To analyze the results of computed tomography of the chest organs, the method of automatic calculation of the volume of the damaged lung tissue according to the ground glass type using the software “Ground glass” (InfoRad 3.0 DICOM Viewer, Moscow, Russia) was used. Segmentation of the right and left lungs and trachea with a threshold of −250 Hounsfield units (HU) was performed. Within the lungs, lesion regions were highlighted with densities in a custom range (default −785 HU to 150 HU). Small vessels that were assumed to be lesions were excluded using a morphological “closure” operation.

For microbiological examination, samples of bronchoalveolar fluid were collected into sterile tubes following aseptic rules. The morning portion of bronchoalveolar fluid was examined. Identification of microorganisms and determination of antibiotic sensitivity were performed on the automated system BD Phoenix-100 (BDBiosciences, San Jose, CA, USA). To assess the taxonomic composition of BAL, a reagent set for DNA isolation from clinical material “RIBO-prep” and reagent sets for detection and quantification of DNA of *Enterobacteriaceae* family, *Pseudomonas aeruginosa*, *Staphylococcus* spp. and *Streptococcus* spp. were used. Qualitative assessment of antibiotic resistance genes was performed using reagent kits for detection of genes of acquired carbapenemases of KPC and OXA-48-like groups (types OXA-48 and OXA-162), genes of acquired carbapenemases of MBL class of VIM, IMP, and NDM groups (Amplisens, Moscow, Russia) by PCR with hybridization-fluorescence detection of amplification products in “real time” mode. The measurements were performed on a CFX 96 Real-Time PCR Detection System (BioRad, Hercules, CA, USA).

## 5. Conclusions

The introduction of adaptive phage therapy in the intensive care units of FRCC ICMR allowed clinicians to apply phage cocktail from the first day of patient admission to the ICU without the use of antibiotics. The elimination of multidrug-resistant microorganisms from the BAL contents and the improvement of lung condition according to CT data, as well as general condition, was achieved. Moreover, a decrease in such inflammatory markers as CRP and procalcitonin was noted. The implementation of the described protocol demonstrates potential as an approach to reduce the number of antibiotics used in intensive care units and maintain their efficacy. Extensive research and large-scale trials are essential to confirm the findings. Moreover, to advance adaptive phage therapy and scale up the approach, registration of a sufficient number of bacteriophage pharmaceutical substances to a wide range of bacteria is required.

## 6. Patents

RU 2 794 585 C2.

## Figures and Tables

**Figure 1 antibiotics-12-01734-f001:**
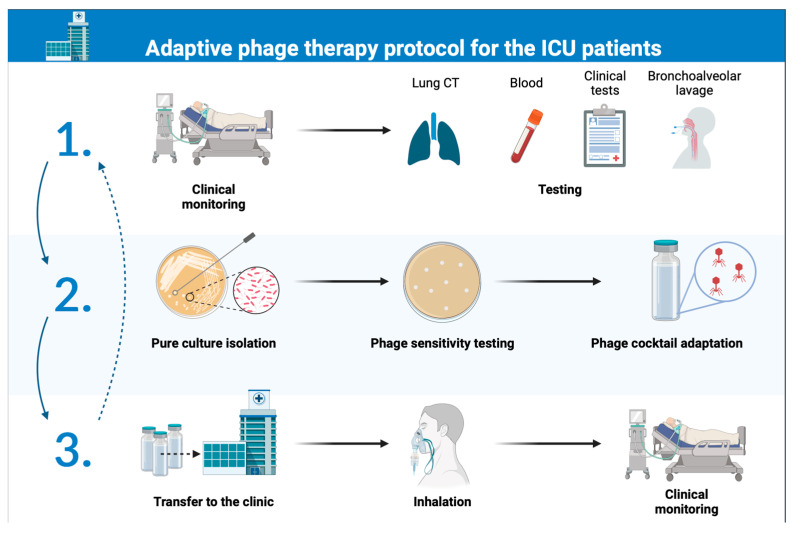
A schematic description of the adaptive phage therapy protocol for the ICU patients; step 1: the medical facility conducts clinical monitoring and collects samples for microbiological monitoring; step 2: the medical facility isolates pure cultures with a minimum of 50 isolates of each target bacterial species, and phage production site conducts sensitivity testing and produces an adapted phage cocktail; step 3: an adapted phage cocktail is transferred to the medical facility for administration to patients under continuous clinical monitoring. CT—computed tomography.

**Table 1 antibiotics-12-01734-t001:** Clinical parameters of the patient 92/23. “-” means no data is available.

Parameters	Results					
	Reference Values	1 Day	7 Days	14 Days	21 Days	28 Days
Bronchoalveolar lavage, PCR	-	1. *K. pneumoniae*—10^5^ CFU/mL; 2. *A. baumannii*—10^5^ CFU/mL	-	1. *K. pneumoniae*—10^6^ CFU/mL; 2. *A. baumannii*—10^6^ CFU/mL	-	No detection
Body temperature, °C	36.6	36.6	36.8	36.7	36.8	36.8
Nature of bronchial secretion, Clinical Pulmonary Infection Score (CPIS)	0—minimal mucous; 1—moderate mucopurulent; 2—purulent	1	1	1	1	1
White blood cells, 10^9^/L	4.5–11	9	7.1	7.7	7.3	7.3
Neutrophils, 10^9^/L	1.9–8.5	5.7	-	-	-	4.4
Platelets, 10^9^/L	152–420	595	-	-	-	461
Lymphocytes, 10^9^/L	1.3–3.1	1.8	-	-	-	1.8
Procalcitonin, ng/mL	<0.1	0.1	0.1	0.1	<0.05	0.1
C-reactive protein (CRP), mg/L	<5	28.65	7.41	10.5	10.7	25.73
Bilirubin, µmol/L	1.7–20	7.3	6.8	6.8	6.9	5
Ventilation mode	-	no	no	no	no	No
Sequential Organ Failure Assessment (SOFA) score	-	2	2	2	2	2

CFU—colony forming units.

**Table 2 antibiotics-12-01734-t002:** Clinical parameters of the patient 62/22. “-” means no data is available.

Parameters	Results					
	Reference Values	1 Day	7 Days	14 Days	21 Days	28 Days
Bronchoalveolar lavage, PCR	-	1. *A. baumannii*—10^4^ CFU/mL; 2. *K. pneumoniae*—10^5^ CFU/mL; 3. *P. aeruginosa*—10^5^ CFU/mL; 4. *S. aureus*—10^5^ CFU/mL	-	1. *P. putida*—10^6^ CFU/mL	-	No detection
Body temperature, °C	36.6	36.6	36.5	36.6	36.7	36.4
Nature of bronchial secretion, Clinical Pulmonary Infection Score (CPIS)	0—minimal mucous; 1—moderate mucopurulent; 2—purulent)	1	1	1	1	0
White blood cells, 10^9^/L	4.5–11	6.9	6.7	6.9	8.9	8.2
Neutrophils, 10^9^/L	1.9–8.5	2.9	-	-	-	3.7
Platelets, 10^9^/L	152–420	156	-	-	-	211
Lymphocytes, 10^9^/L	1.3–3.1	2.7	-	-	-	2.8
Procalcitonin, ng/mL	<0.1	<0.05	<0.05	<0.05	<0.05	<0.05
C-reactive protein (CRP), mg/L	<5	29.25	20.13	23.75	19.87	11.5
Bilirubin, µmol/L	1.7–20	8.8	7.7	6.4	5.7	6.5
Ventilation mode	-	no	no	no	no	no
Sequential Organ Failure Assessment (SOFA) score	-	3	5	4	4	1

**Table 3 antibiotics-12-01734-t003:** Clinical parameters of the patient 847/21. “-” means no data is available.

Parameters	Results					
	Reference Values	1 Day	7 Days	14 Days	21 Days	28 Days
Bronchoalveolar lavage, PCR	-	1. *A. baumannii*—10^5^ CFU/mL; 2. *K. pneumoniae*—10^4^ CFU/mL	-	-	-	No detection
Body temperature, °C	36.6	37.8	37.3	36.7	36.6	36.8
Nature of bronchial secretion, Clinical Pulmonary Infection Score (CPIS)	0—minimal mucous; 1—moderate mucopurulent; 2—purulent)	1	1	1	1	0
White blood cells, 10^9^/L	4.5–11	10.5	9.4	10.7	13.6	10.9
Neutrophils, 10^9^/L	1.9–8.5	8.0	-	-	-	8.4
Platelets, 10^9^/L	152–420	311	-	-	-	416
Lymphocytes, 10^9^/L	1.3–3.1	1.5	-	-	-	1.9
Procalcitonin, ng/mL	<0.1	0.5	-	-	-	0.1
C-reactive protein (CRP), mg/L	<5	150	90.5	77.9	99.6	28.2
Bilirubin, µmol/L	1.7–20	27.4	36.3	29.9	26.4	19
Ventilation mode	-	BIPAP	BIPAP	BIPAP	CPAP	CPAP
Sequential Organ Failure Assessment (SOFA) score	-	4	5	4	4	4

BIPAP—biphasic positive airway pressure; CPAP—continuous positive airway pressure.

## Data Availability

Data are contained within the article and Appendix A.

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
