# Peer review of "Adaptive Phage Therapy for the Prevention of Recurrent Nosocomial Pneumonia: Novel Protocol Description and Case Series"

_antibiotics, 2023, doi:10.3390/antibiotics12121734_

Round 1
Reviewer 1 Report
Comments and Suggestions for Authors
I consider this manuscript suitable for publication but only after the authors address the following minor issues.
Minor issues:
- Line 26: please write what CRP means.
- Lines 126: please write what ICU and FRCC ICMR means.
- Line 130: please explain what SAPS II score is.
- Line 140: please write what ICU means.
- Line 163: please write what HU means.
- Line 199: please write what RPC “Micromir” means.
- Line 91: please write what CPIS score is.
Author Response
We thank the reviewer for his time and for providing revisions. All corrections have been made. Point-by-point response:
Minor issues:
- Line 26: please write what CRP means.
Response: clarified.
- Lines 126: please write what ICU and FRCC ICMR means.
Response: clarified.
- Line 130: please explain what SAPS II score is.
Response: clarified.
- Line 140: please write what ICU means.
Response: clarified.
- Line 163: please write what HU means.
Response: clarified.
- Line 199: please write what RPC “Micromir” means.
Response: clarified.
- Line 91: please write what CPIS score is.
Response: clarified.
Reviewer 2 Report
Comments and Suggestions for Authors
The article, "Adaptive Phage Therapy: Protocol and First Clinical Cases," outlines the protocol for clinical use and explores the potential for future implementation. The rising interest in employing bacteriophages to combat antibiotic-resistant bacteria through adaptive phage therapy protocols in intensive care units is emphasized, shedding light on the challenges posed by regulatory requirements that impede commercial development. The study presents clinical cases where patients received customized bacteriophage conjugates, showcasing the potential to reduce antibiotic use and effectively combat multidrug-resistant microorganisms. Nevertheless, the article underscores the necessity for further research and regulatory adjustments to fully unlock the benefits of adaptive phage therapy in intensive care units.
Suggestions for improving the article:
Introduction:
- The introduction provides a comprehensive overview, consider refining the research questions and objectives, specifically addressing the regulatory framework.
- To strengthen the validity of the results, mention the absence of a control group in the clinical case. Comparisons with standard antibiotic treatment groups or a placebo control would enhance the study.
Materials and Methods:
- Include patient demographics such as age, gender, and comorbidities in the Materials and Methods section for a better understanding of the patient population.
Results:
- The results are promising, incorporate statistical analysis, including p-values or confidence intervals, to ascertain the significance of observed changes.
Discussion:
- Provide a more critical analysis of study limitations, considering potential confounders and addressing the small sample size for a balanced interpretation.
- Discuss the long-term implications and challenges of implementing adaptive phage therapy at scale, including regulatory, logistical, and ethical considerations.
Conclusion: Fine but can be improved!
- Need for further research and large-scale studies to validate results. Highlight limitations and potential challenges in implementing this approach.
Citation Recommendations:
Abdelrahman, F., Easwaran, M., Daramola, O., Raghav, S., Lynch, S., Odussel, T. J., Khan, F. M., Ayobami, A., Adnan, F., Trent, E., et al. (2021). Phage-encoded endolysin. Antibiotics, 10(2), 124. https://doi.org/10.3390/antibiotics10020124
Sanmuk, S. G., Admela, J., Moya-Anderico, L., Feher, T., Arevalo-James, B. V., Blanco-Cabra, N., Torrents, E. (2023). Phage Therapy: Assessing the in vivo efficacy of clinically isolated phages against uropathogenic and invasive biofilm-forming Escherichia coli strains. Cells, 12(3), 344. https://doi.org/10.3390/cells12030344
Comments on the Quality of English LanguageThe English language quality is fine except for some minor spell check!
Author Response
Our deepest gratitude to the reviewer for his suggestions. We have taken into account all comments and made corrections to the manuscript. Point-by-point response:
Suggestions for improving the article:
Introduction:
- The introduction provides a comprehensive overview, consider refining the research questions and objectives, specifically addressing the regulatory framework.
Response: We are grateful to the reviewer for his advice. We have added up-to-date information on the number of individual phage therapy cases and a brief information on the protocol and case series. Otherwise, the article refers to the most current data in the regulation of phage therapy.
- To strengthen the validity of the results, mention the absence of a control group in the clinical case. Comparisons with standard antibiotic treatment groups or a placebo control would enhance the study.
Response: Mentioned it as one of the limitations in the Discussion section.
Materials and Methods:
- Include patient demographics such as age, gender, and comorbidities in the Materials and Methods section for a better understanding of the patient population.
Response: Included in the M&M section.
Results:
- The results are promising, incorporate statistical analysis, including p-values or confidence intervals, to ascertain the significance of observed changes.
Response: Thank you for your thoughtful comments and suggestions regarding our manuscript, specifically the request to calculate p-values comparing the data from the patients. We appreciate your input and the opportunity to discuss the methodology of our study. We agree that statistical analysis is crucial in research; however, in the context of our study, which presents three case reports, calculating p-values may not be appropriate. The main reasons are:
Small Sample Size: With only three cases, the statistical power is extremely limited, making any p-value calculation unreliable and potentially misleading.
Nature of Case Reports: Our study is descriptive and exploratory, focusing on individual clinical experiences. Case reports typically do not provide a basis for statistical hypothesis testing but are used to generate hypotheses for future research.
Given these considerations, we believe that including p-values could lead to misinterpretation. We aim to contribute to the field by highlighting unique findings and suggesting directions for future.
We appreciate your understanding and are open to further suggestions to improve our manuscript.
Discussion:
- Provide a more critical analysis of study limitations, considering potential confounders and addressing the small sample size for a balanced interpretation.
Response: Provided.
- Discuss the long-term implications and challenges of implementing adaptive phage therapy at scale, including regulatory, logistical, and ethical considerations.
Response: Provided.
Conclusion: Fine but can be improved!
- Need for further research and large-scale studies to validate results. Highlight limitations and potential challenges in implementing this approach.
Response: Provided.
Citation Recommendations:
Abdelrahman, F., Easwaran, M., Daramola, O., Raghav, S., Lynch, S., Odussel, T. J., Khan, F. M., Ayobami, A., Adnan, F., Trent, E., et al. (2021). Phage-encoded endolysin. Antibiotics, 10(2), 124. https://doi.org/10.3390/antibiotics10020124
Sanmuk, S. G., Admela, J., Moya-Anderico, L., Feher, T., Arevalo-James, B. V., Blanco-Cabra, N., Torrents, E. (2023). Phage Therapy: Assessing the in vivo efficacy of clinically isolated phages against uropathogenic and invasive biofilm-forming Escherichia coli strains. Cells, 12(3), 344. https://doi.org/10.3390/cells12030344
Response: We thank the reviewer for advice on citations. It seems to us that they do not exactly match the topic of the present paper, as they focus more on phage proteins and in vivo testing in in the Galleria Mellonella infection model. In the present paper we wanted to focus on aspects of phage therapy application in humans and regulatory issues.
Reviewer 3 Report
Comments and Suggestions for Authors
Zurabov et al. describe the prophylactic and therapeutic application of a broad host range, multi-phage cocktail to patients in an intensive care unit to prevent or treat hospital acquired pneumonia. The study is ground breaking as it not only shows the safety and efficacy of bacteriophage therapy for individual patients but also outlines a therapeutic strategy, the authors call “adaptive phage therapy” to implement phage therapy in the clinic. The study should be published with urgency and this reviewer would like to see the manuscript to receive all the promotion possible to reach a wide distribution.
Specific comments:
Material and Methods, lines 142-147: From this paragraph it is not entirely clear how many bacteriophages the cocktail actually contained. Were there 3-4 phages against each of the 14 clinical strains listed, so 42-56 phages in total? From the text it could be concluded that 3-4 phages would be sufficient to cover all the clinical strains mentioned. In light of the diversity of the indicated pathogens that would not be realistic. The authors should clarify this point. P.S. The reviewer noted that the authors clarified the point in the discussion, however, the text in the Material and Methods section should still be adjusted.
Results, line 228 and also later in the text: The sentence “The rest of the media were sterile.” is unclear in this context. Do the authors mean that “No other pathogenic bacteria were detected in the investigated samples.”?
Discussion, lines 286-286: The sentence “This process is a (delete the "a") very challenging to scale up, leading to a loss of time that can be lifesaving for some patients.” is logically incorrect. The loss of time is not lifesaving, but life threatening. The option to start phage therapy immediately after hospitalization without patient-specific adjustments is lifesaving.
Author Response
Our deepest gratitude to the reviewer for his appreciation of our work. We have taken into account all comments and made corrections to the manuscript. Point-by-point response:
Specific comments:
Material and Methods, lines 142-147: From this paragraph it is not entirely clear how many bacteriophages the cocktail actually contained. Were there 3-4 phages against each of the 14 clinical strains listed, so 42-56 phages in total? From the text it could be concluded that 3-4 phages would be sufficient to cover all the clinical strains mentioned. In light of the diversity of the indicated pathogens that would not be realistic. The authors should clarify this point. P.S. The reviewer noted that the authors clarified the point in the discussion, however, the text in the Material and Methods section should still be adjusted.
Response: clarified in the M&M section.
Results, line 228 and also later in the text: The sentence “The rest of the media were sterile.” Is unclear in this context. Do the authors mean that “No other pathogenic bacteria were detected in the investigated samples.”?
Response: rephrased the sentence.
Discussion, lines 286-286: The sentence “This process is a (delete the “a”) very challenging to scale up, leading to a loss of time that can be lifesaving for some patients.” Is logically incorrect. The loss of time is not lifesaving, but life threatening. The option to start phage therapy immediately after hospitalization without patient-specific adjustments is lifesaving.
Response: corrected.
Reviewer 4 Report
Comments and Suggestions for Authors
I am grateful for the chance to review the manuscript titled "Adaptive phage therapy: protocol and first clinical cases" for potential publication in Antibiotic. Although the manuscript is intriguing, I have substantial concerns that must be addressed to consider it for publication. I sincerely hope that the recommendations provided will prove valuable to the authors in their endeavor to publish this work.
It is worth considering that the manuscript's inclusion of only three clinical cases may make it more suitable for a short communication rather than a full article. To enhance the robustness of the findings regarding the effectiveness of bacteriophage complex via inhalation, future studies could benefit from a larger sample size.
To ensure the work remains relevant and suitable for a short communication, it is recommended to include more recent references in the manuscript. While the current references provide support for the statements made, it is worth noting that the manuscript has been cited in 7 out of the 21 most recent publications within the past five years. Including additional recent citations would further strengthen the manuscript's significance and demonstrate its continued impact in the field.
The manuscript under review appears to have a similar study protocol and ethics approval number as a previous publication titled "Adaptive Phage Therapy in the Treatment of Patients with Recurrent Pneumonia (Pilot Study)" (https://doi.org/10.15360/1813-9779-2021-6-4-14). However, there are some differences in the inclusion and exclusion criteria used in the two studies. To ensure transparency and avoid redundancy in the scientific literature, it would be beneficial for the author to provide further clarification and justification for presenting similar findings in both studies.
The manuscript title could benefit from a more specific and comprehensive wording to accurately convey the novelty and significance of the study's findings.
The introduction section of the manuscript provides a comprehensive overview of the challenges posed by antibiotic resistance and the potential of phage therapy. However, it may benefit from focusing more specifically on the objective of the study and providing a concise summary of the protocol and clinical cases.
The manuscript should provide more detailed information and appropriate citations regarding the preparation of the phage cocktail used in the study. This will help readers understand the composition and methodology of the phage therapy.
The method section of the manuscript lacks information about blood gas measurements and blood chemistry profiles, which are important for evaluating the patients' physiological status. Additionally, the high platelet count in patient serum raises questions about the severity of the condition and its correlation with the high Sequential Organ Failure Assessment (SOFA) score. Further clarification and justification from the author regarding these discrepancies would enhance the understanding of the study findings.
I would suggest that the manuscript includes the antibiotic susceptibility test results for all bacterial isolates in the results section. This would provide additional evidence and support the claim of multidrug-resistant microorganism elimination through the inhalation of the bacteriophage complex, enhancing the scientific rigor of the study.
The first time a new species is mentioned, it is necessary to spell out the full genus name. However, in subsequent mentions of the species, the genus should be abbreviated. Additionally, it is important to ensure that all scientific names are properly italicized. Please ensure to thoroughly review and verify the accuracy of the bacterial names throughout the entire manuscript.
I would recommend that the author thoroughly addresses the study's limitations within the discussion section. By acknowledging and discussing the limitations, the author can provide a more comprehensive and transparent interpretation of the findings, enhancing the overall quality of the manuscript.
Author Response
I am grateful for the chance to review the manuscript titled "Adaptive phage therapy: protocol and first clinical cases" for potential publication in Antibiotic. Although the manuscript is intriguing, I have substantial concerns that must be addressed to consider it for publication. I sincerely hope that the recommendations provided will prove valuable to the authors in their endeavor to publish this work.
Response: We are grateful to the reviewer for his valuable comments. We have taken all suggestions into account.
It is worth considering that the manuscript's inclusion of only three clinical cases may make it more suitable for a short communication rather than a full article. To enhance the robustness of the findings regarding the effectiveness of bacteriophage complex via inhalation, future studies could benefit from a larger sample size.
Response: The main value of the manuscript is the first described protocol of interaction between phage center and clinical institution, the cases presented here are the primary confirmation of the feasibility of this protocol, no doubt we are going to continue the work and study our protocol in larger cohorts with experimental and control groups. It seems to us that the paper is worthy of publication as a separate article rather than a short communication. We have also received confirmation from phage therapy specialists that this is a highly relevant topic in the field.
To ensure the work remains relevant and suitable for a short communication, it is recommended to include more recent references in the manuscript. While the current references provide support for the statements made, it is worth noting that the manuscript has been cited in 7 out of the 21 most recent publications within the past five years. Including additional recent citations would further strengthen the manuscript's significance and demonstrate its continued impact in the field.
Response: We have added up-to-date information on the number of individual phage therapy cases. Otherwise, the article refers to the most current data in the regulation of phage therapy. 10 of the 22 citations refer to articles published from 2019 onwards. 14 out of 22 refer to articles published from 2018 onwards.
The manuscript under review appears to have a similar study protocol and ethics approval number as a previous publication titled "Adaptive Phage Therapy in the Treatment of Patients with Recurrent Pneumonia (Pilot Study)" (https://doi.org/10.15360/1813-9779-2021-6-4-14). However, there are some differences in the inclusion and exclusion criteria used in the two studies. To ensure transparency and avoid redundancy in the scientific literature, it would be beneficial for the author to provide further clarification and justification for presenting similar findings in both studies.
Response: The decision of the Ethics Committee authorizes the use of adaptive phage therapy in intensive care units in different categories of patients. In order to expand the possibilities of using adaptive phage therapy and for a comprehensive study of its possible effects, in addition to the already published pilot study, a scientific research and registration of cases of adaptive phage therapy technology in patients for whom inclusion in the pilot study was not possible was carried out. The data obtained may be useful for adjusting the final design of a future large-scale clinical trial.
The manuscript title could benefit from a more specific and comprehensive wording to accurately convey the novelty and significance of the study's findings.
Response: reworked the title.
The introduction section of the manuscript provides a comprehensive overview of the challenges posed by antibiotic resistance and the potential of phage therapy. However, it may benefit from focusing more specifically on the objective of the study and providing a concise summary of the protocol and clinical cases.
Response: Added information to the Introduction.
The manuscript should provide more detailed information and appropriate citations regarding the preparation of the phage cocktail used in the study. This will help readers understand the composition and methodology of the phage therapy.
Response: Added more information to the M&M section. It seems to us that it is not necessary to overwhelm the materials and methods section with an exceedingly detailed description of phage production, as all the steps are standardized enough. The peculiarity is that each phage is produced separately, which we pointed out.
The method section of the manuscript lacks information about blood gas measurements and blood chemistry profiles, which are important for evaluating the patients' physiological status. Additionally, the high platelet count in patient serum raises questions about the severity of the condition and its correlation with the high Sequential Organ Failure Assessment (SOFA) score. Further clarification and justification from the author regarding these discrepancies would enhance the understanding of the study findings.
Response: A high platelet count does not affect the SOFA score. The main contributors to the SOFA score in patient 92/23 was the level of consciousness. In patient 62/22, the patient's level of consciousness, which was a minimum of 8 on the Glasgow Coma Scale, and the episode of low blood pressure were the main contributors to the SOFA score. In patient 847/21, the high SOFA was due to the patient's level of consciousness and the bilirubin level. We have added the information about bilirubin levels and comorbidities.
I would suggest that the manuscript includes the antibiotic susceptibility test results for all bacterial isolates in the results section. This would provide additional evidence and support the claim of multidrug-resistant microorganism elimination through the inhalation of the bacteriophage complex, enhancing the scientific rigor of the study.
Response: Added Tables to the Supplementary materials.
The first time a new species is mentioned, it is necessary to spell out the full genus name. However, in subsequent mentions of the species, the genus should be abbreviated. Additionally, it is important to ensure that all scientific names are properly italicized. Please ensure to thoroughly review and verify the accuracy of the bacterial names throughout the entire manuscript.
Response: Verified.
I would recommend that the author thoroughly addresses the study's limitations within the discussion section. By acknowledging and discussing the limitations, the author can provide a more comprehensive and transparent interpretation of the findings, enhancing the overall quality of the manuscript.
Response: Indicated the limitations of the study in the discussion section and adjusted the conclusions.
Round 2
Reviewer 2 Report
Comments and Suggestions for Authors
The article after revision is complete and acceptable.
I still would like to ask the authors to reconsider the addition of a second reference as it aligns well with our manuscript's scope. It addresses endotoxin removal for phages (a limiting factor for invivo bacteriophage therapy) and phage-bound depolymerase activity against biofilms of nonspecific strains in vivo. We believe the inclusion of this reference will indeed strengthen the argument for regulatory approvals of phage therapy in humans!
Author Response
The article after revision is complete and acceptable.
I still would like to ask the authors to reconsider the addition of a second reference as it aligns well with our manuscript's scope. It addresses endotoxin removal for phages (a limiting factor for invivo bacteriophage therapy) and phage-bound depolymerase activity against biofilms of nonspecific strains in vivo. We believe the inclusion of this reference will indeed strengthen the argument for regulatory approvals of phage therapy in humans!
Sanmuk, S. G., Admela, J., Moya-Anderico, L., Feher, T., Arevalo-James, B. V., Blanco-Cabra, N., Torrents, E. (2023). Phage Therapy: Assessing the in vivo efficacy of clinically isolated phages against uropathogenic and invasive biofilm-forming Escherichia coli strains. Cells, 12(3), 344. https://doi.org/10.3390/cells12030344
Response: We are grateful to the reviewer for advising us to include this publication to strengthen our manuscript. We have again analyzed the proposed article and, despite its great contribution to the field of phage therapy, we believe that it is not really suitable for inclusion in our publication as it does not address regulatory issues but reports results based on phage testing on Galleria mellonella. Undoubtedly, phages must be purified before administration. Moreover, endotoxin release during bacterial lysis in vivo is considered a potential problem that cannot be solved by purification of phage preparations because lysis occurs at the site of infection, but complete data on endotoxin release and its effects are rare and inconsistent (Liu D, Van Belleghem JD, de Vries CR, et al. The Safety and Toxicity of Phage Therapy: A Review of Animal and Clinical Studies. Viruses. 2021;13(7):1268. doi:10.3390/v13071268). Meanwhile, data from clinical trials and clinical cases show high safety and good tolerability of phage preparations, as we mentioned in the introduction section.
Reviewer 4 Report
Comments and Suggestions for Authors#1 I appreciate the author's efforts in addressing several of the concerns raised. I will continue to engage with the author to ensure that the remaining issues are thoroughly discussed and resolved to improve the overall quality and impact of the manuscript.
#2 I value the perspective offered on the manuscript's significance and the affirmation from phage therapy specialists. Nevertheless, I maintain the view that the pioneering protocol of phage therapy in a clinical setting, along with the confirmation of relevance, underscores the importance of categorizing this work as a communication. It may be beneficial to discuss with the editor the possibility of specifying the classification as a "case series" to offer additional clarity regarding the content's nature.
#5 Thank you for revising the title to "Adaptive phage therapy: novel protocol description and case series." While the revised title captures the inclusion of a novel protocol and case series, I still believe that a more specific indication of the targeted condition, such as sepsis or septic shock, would enhance the clarity and relevance of the study's focus. I appreciate your attention to this matter and look forward to reviewing the revised manuscript.
#9 Thank you for providing the additional information in the supplementary materials. However, I would like to initiate a discussion regarding the interpretation of the antibiotic susceptibility test results for the bacterial isolates. The table demonstrates that all bacteria classified as resistant to the Extended-spectrum beta-lactamase (ESBL) were also resistant to Amoxicillin/Clavulanate. The MIC of each bacterial isolate is as follows: K. pneumoniae from patient 92/23 (>32/2 mg/l), patient 62/22 (>16/2 mg/l), and patient 847/21 (>32/2 mg/l).
Given that traditional ESBLs are inhibited by all β-lactamase inhibitors, including clavulanate. European Committee on Antimicrobial Susceptibility Testing (EUCAST) recommends using a fixed concentration of 2 mg/L clavulanate for susceptibility testing of the amoxicillin/clavulanate combination, whereas the Clinical and Laboratory Standards Institute (CLSI) recommends using a 2:1 ratio of amoxicillin/clavulanate. Additionally, CLSI considers Enterobacteriaceae with MICs 16/8 mg/L as intermediate, and those with MICs >32/16 mg/L as resistant. A thorough discussion on the rationale behind the choice of interpretation criteria would enhance the scientific rigor of the study.
Author Response
#1 I appreciate the author's efforts in addressing several of the concerns raised. I will continue to engage with the author to ensure that the remaining issues are thoroughly discussed and resolved to improve the overall quality and impact of the manuscript.
Response: We would like to express our deepest gratitude to the reviewer for his time and effort in improving our manuscript. We have answered all the remaining questions and are confident that by working together we have greatly improved the quality of the publication.
#2 I value the perspective offered on the manuscript's significance and the affirmation from phage therapy specialists. Nevertheless, I maintain the view that the pioneering protocol of phage therapy in a clinical setting, along with the confirmation of relevance, underscores the importance of categorizing this work as a communication. It may be beneficial to discuss with the editor the possibility of specifying the classification as a "case series" to offer additional clarity regarding the content's nature.
Response: We are thankful to the reviewer for his valuable opinion on the format of the publication. We propose to leave this point to the decision of the editor. The authors of the manuscript are of the opinion that the work is suitable for publication as a separate article.
#5 Thank you for revising the title to "Adaptive phage therapy: novel protocol description and case series." While the revised title captures the inclusion of a novel protocol and case series, I still believe that a more specific indication of the targeted condition, such as sepsis or septic shock, would enhance the clarity and relevance of the study's focus. I appreciate your attention to this matter and look forward to reviewing the revised manuscript.
Response: We express our gratitude to the reviewer for his advice on improving the title of the article. We have added the designation of the targeted condition to the title.
#9 Thank you for providing the additional information in the supplementary materials. However, I would like to initiate a discussion regarding the interpretation of the antibiotic susceptibility test results for the bacterial isolates. The table demonstrates that all bacteria classified as resistant to the Extended-spectrum beta-lactamase (ESBL) were also resistant to Amoxicillin/Clavulanate. The MIC of each bacterial isolate is as follows: K. pneumoniae from patient 92/23 (>32/2 mg/l), patient 62/22 (>16/2 mg/l), and patient 847/21 (>32/2 mg/l).
Given that traditional ESBLs are inhibited by all β-lactamase inhibitors, including clavulanate. European Committee on Antimicrobial Susceptibility Testing (EUCAST) recommends using a fixed concentration of 2 mg/L clavulanate for susceptibility testing of the amoxicillin/clavulanate combination, whereas the Clinical and Laboratory Standards Institute (CLSI) recommends using a 2:1 ratio of amoxicillin/clavulanate. Additionally, CLSI considers Enterobacteriaceae with MICs 16/8 mg/L as intermediate, and those with MICs >32/16 mg/L as resistant. A thorough discussion on the rationale behind the choice of interpretation criteria would enhance the scientific rigor of the study.
Response: We thank the reviewer for his careful and thoughtful consideration of this subject. In the microbiological laboratory of the FRCC ICMR identification and antibiotic sensitivity determination is performed on an automatic bacteriological analyzer BD Phoenix-100 (BDBiosciences, USA), the data are interpreted according to the EUCAST protocols uploaded to the analyzer. We fully agree with you that the use of CLSI standards for protected β-lactam antibiotics (Amoxiclav) will increase the scientific significance of the study, as it will allow to evaluate different ratios of the antimicrobial drug and beta-lactamase inhibitor. We will do our best to take your comment into account when conducting the study in the future. We have added additional information and references to the limitations in the Discussion section.
Round 3
Reviewer 4 Report
Comments and Suggestions for Authors
Thank you for your response, and I appreciate your efforts in addressing the concerns raised. I'm glad to see that you have taken my comments into consideration, and I hope that the suggested improvements will further enhance the quality and scientific rigor of your study.